



# Development of a small unmanned aircraft system to derive $CO_2$ emissions of anthropogenic point sources

Maximilian Reuter[1], Heinrich Bovensmann[1], Michael Buchwitz[1], Jakob Borchardt[1], Sven Krautwurst[1], Konstantin Gerilowski[1], Matthias Lindauer[2], Dagmar Kubistin[2], and John P. Burrows[1]

[1]University of Bremen, Institute of Environmental Physics, Bremen, Germany
[2]Deutscher Wetterdienst, Meteorologisches Observatorium Hohenpeißenberg, Hohenpeißenberg, Germany

**Correspondence:** Maximilian Reuter (mail@maxreuter.org)

**Abstract.** A reduction of the anthropogenic emissions of $CO_2$ (carbon dioxide) is necessary to stop or slow down man-made climate change. To verify mitigation strategies, a global monitoring system such as the envisaged European Copernicus anthropogenic $CO_2$ monitoring mission (CO2M) is required. Those satellite data are going to be complemented and validated with airborne measurements. UAV (unmanned aerial vehicle) based measurements can provide a cost-effective way to contribute to these activities. Here we present the development of a sUAS (small unmanned aircraft system) to quantify the $CO_2$ emissions of a nearby point source from its downwind mass flux without the need for any ancillary data. Specifically, $CO_2$ is measured by a NDIR (non-dispersive infrared) detector and the wind speed and direction is measured with a 2D ultrasonic acoustic resonance anemometer. By means of laboratory measurements and an in-flight validation at the ICOS (Integrated Carbon Observation System) atmospheric station Steinkimmen (STE) near Bremen, Germany, we estimate that the individual $CO_2$ measurements have a precision of 3 ppm and that $CO_2$ enhancements can be determined with an accuracy of 1.3% or 0.9 ppm, whichever is larger. We introduce an anemometer calibration method to minimize the effect of rotor downwash on the wind measurements. This method derives the fit parameters of a linear calibration model accounting for scaling, rotation, and a potential constant bias. For this purpose it analyzes wind measurements taken while following a suitable flight pattern and assuming stationary wind conditions. From the calibration and validation experiments, we estimate the single measurement precision of the horizontal wind speed to be $0.40\,\mathrm{m\,s^{-1}}$ and the accuracy to be $0.33\,\mathrm{m\,s^{-1}}$. By means of two flights downwind of the ExxonMobil natural gas processing facility in Großenkneten about 40 km east of Bremen, Germany, we demonstrate how the measurements of elevated $CO_2$ concentrations can be used to infer mass fluxes of atmospheric $CO_2$ related to the emissions of the facility.

## 1 Introduction

$CO_2$ (carbon dioxide) emissions are the primary cause of man-made climate change and in order to limit this, a reduction of emissions is necessary (IPCC, 2013). Large parts of the anthropogenic $CO_2$ emissions originate from point sources such as coal or gas fired power plants and observing systems are needed to verify mitigation strategies through independent measurements (Pinty et al., 2017). On a global level, it is planned to monitor $CO_2$ emissions remotely by means of satellites (e.g., Nassar et al., 2017; Reuter et al., 2019) such as the envisaged European Copernicus anthropogenic $CO_2$ monitoring mission (CO2M,





Janssens-Maenhout et al., 2020) building upon the heritage of CarbonSat (Bovensmann et al., 2010; Buchwitz et al., 2013),
which was a candidate for ESA's Earth Explorer-8 mission. At the regional level, this can be achieved by a dense network of
ground based observations or by airplane based measurements, which can also be used for smaller point sources and for the
validation of the satellite data (e.g., Krings et al., 2011, 2018; Carotenuto et al., 2018). UAV (unmanned aerial vehicle) based
measurements can complement these activities, reduce flight costs, and enable measurements under conditions not suitable for
satellite measurements, e.g., in cloud contaminated scenes, during night, or at facilities with emissions below the satellite's
detection limit.

The quantification of emissions with the mass balance approach using atmospheric measurements downwind of the source
requires knowledge of the atmospheric concentration of $CO_2$, the wind speed, and the density of air. As done, e.g., by Krings
et al. (2018), the $CO_2$ enhancement from the source can be computed from the measurements of a cross-section of its plume by
subtracting the $CO_2$ background concentration derived from measurements outside the plume. Additionally, measurements of
the wind speed and the density of air are needed to compute the $CO_2$ cross-sectional flux by integrating over the cross-sectional
flux density.

Earlier studies showed that UAVs are suitable to serve as platforms for in situ $CO_2$ sensors (e.g., Berman et al., 2012; Khan
et al., 2012; Kunz et al., 2018; Allen et al., 2019; Chiba et al., 2019), anemometers (e.g., Palomaki et al., 2017; Hollenbeck
et al., 2018; Shimura et al., 2018; Barbieri et al., 2019), and/or sensors for meteorological parameters (e.g., Barbieri et al.,
2019). Our goal was to create a sUAS (small unmanned aircraft system) which is capable to determine atmospheric $CO_2$ mass
fluxes from its own sensor data independently from external data sources. The system was required to be reliable, affordable,
and to be realized within a relatively short development phase. For this reason we decided to use only commercially available
and mature components for the UAV as well sensor equipment.

In the following section, we introduce the hardware setup and the instrumentation. In Sec. 3 we characterize the measure-
ments of our in situ $CO_2$ sensor by comparing them with those from an accurate laboratory instrument and in Sec. 4 we
introduce a calibration method for the anemometer correcting, e.g., for the influence of rotor downwash. In Sec. 5 we quantify
the performance of our atmospheric measurements with focus on $CO_2$ and wind by analyzing two validation flights at the ICOS
(Integrated Carbon Observation System) atmospheric station Steinkimmen (STE) near Bremen, Germany. In Sec. 6 we demon-
strate how the sUAS can be used to measure elevated $CO_2$ concentrations from a nearby source and how these measurements
can be used to infer the $CO_2$ mass flux related to the emissions of the source. Finally, we summarize and conclude our results.

## 2 Hardware setup and instrumentation

We use a DJI Matrice 210v2 (*https://www.dji.com*, last access 02.06.2020) as UAV airborne plattform, which weights about
4.8 kg (including batteries) and can carry a maximum payload of 1.34 kg. An overview of the sUAS including UAV and
instrumentation is shown by Fig. 1. The DJI Matrice 210v2 is powered by two LiPo (Lithium Polymer) batteries with a capacity
of 7660 mAh at 22.8 V allowing for flight times typically in the range of 24 to 34 minutes, depending on payload. It is optimized





and often used for industrial applications and emergency services. Its top cover has four mounting threads designed to attach additional payloads and we use it to mount an aluminum base plate.

The Vaisala GMP343 in situ $CO_2$ sensor (*https://www.vaisala.com*, last access 02.06.2020) is the primary measurement device aboard the UAV and the only part of the equipment not attached to the base plate but to the secondary gimbal connector

of the UAV. The GMP343 has a NDIR (non-dispersive infrared) sensor that uses an electrically tunable Fabry-Perot interferometer to switch back and forth between a $CO_2$ absorption band at around $4.26\,\mu m$ and the neighboring continuum at around $3.9\,\mu m$ serving as reference. It is specified to measure the mole fraction of atmospheric $CO_2$ with a single sounding precision of $\pm 3$ ppm ($1\sigma$). In order to achieve a fast response time, we operate the $CO_2$ sonde with an open measuring cell through which the ambient air flows.

The second most important sensor aboard the UAV is the FT Technologies FT205 anemometer (*https://fttechnologies.com*, last access 02.06.2020). It is a lightweight 2D ultrasonic acoustic resonance anemometer specifically designed for UAV applications. It is specified to deliver measurements of the horizontal wind speed with a precision of $\pm 0.3\,\mathrm{m\,s^{-1}}$ for wind speeds below $16\,\mathrm{m\,s^{-1}}$. In order to reduce the influence of the rotor downwash on the wind measurements, we mounted the anemometer $36\,\mathrm{cm}$ above the rotor plane on a carbon fiber pole with a diameter of $7\,\mathrm{mm}$.

Deriving the $CO_2$ mass flux from the atmospheric mole fraction and the wind speed requires knowledge of the molar air density, which we compute from the measured meteorological parameters. For this purpose we use two Adafruit (*https://www.adafruit.com*, last access 02.06.2020) breakout boards equipped with a Bosch BMP388 pressure sensor and a Sensirion SHT31-DIS temperature and humidity sensor. The BMP388 pressure sensor has a specified absolute accuracy of $\pm 0.5\,\mathrm{hPa}$ ($\pm 0.08\,\mathrm{hPa}$ relative accuracy). The Sensirion SHT31-DIS temperature and relative humidity measurements have a

specified typical accuracy of $\pm 0.3°C$ and $\pm 2\%$, respectively. GNSS (global navigation satellite system) position as well as attitude information is provided in-flight directly by the UAV via DJI's OSDK API (onboard software development kit application programming interface).

With a maximum rate of $0.5\,\mathrm{Hz}$, the $CO_2$ sensor offers the slowest readout rates of our sensor equipment and we decided to synchronize all instrument readouts with those of the GMP343 in order to generate a convenient output data stream with the

same time-stamp for all sensors.

A Dronee Zoon v2 telemetry module (*https://dronee.aero*, last access 02.06.2020) that allows transmission ranges up to $3\,\mathrm{km}$ establishes an independent bidirectional data-link. It is used to transmit the most important measurement data ($CO_2$, wind, position, etc.) to the ground enabling, e.g., to optimize the flight pattern according to the wind measurements during flight.

All instruments are connected to a Raspberry Pi 3B+ via a RS-232 to USB converter ($CO_2$), a RS-485 to USB converter (wind), a TTL UART to USB converter (UAV), USB cable (telemetry), and $I^2C$-bus (pressure, temperature, and rel. humidity). The Raspberry Pi as well as the entire science payload is powered by the batteries of the UAV.

Beside the science payload, the UAV carries also a DJI X4S camera for documentary purposes and for high quality, gimbal-stabilized first person view during flight.





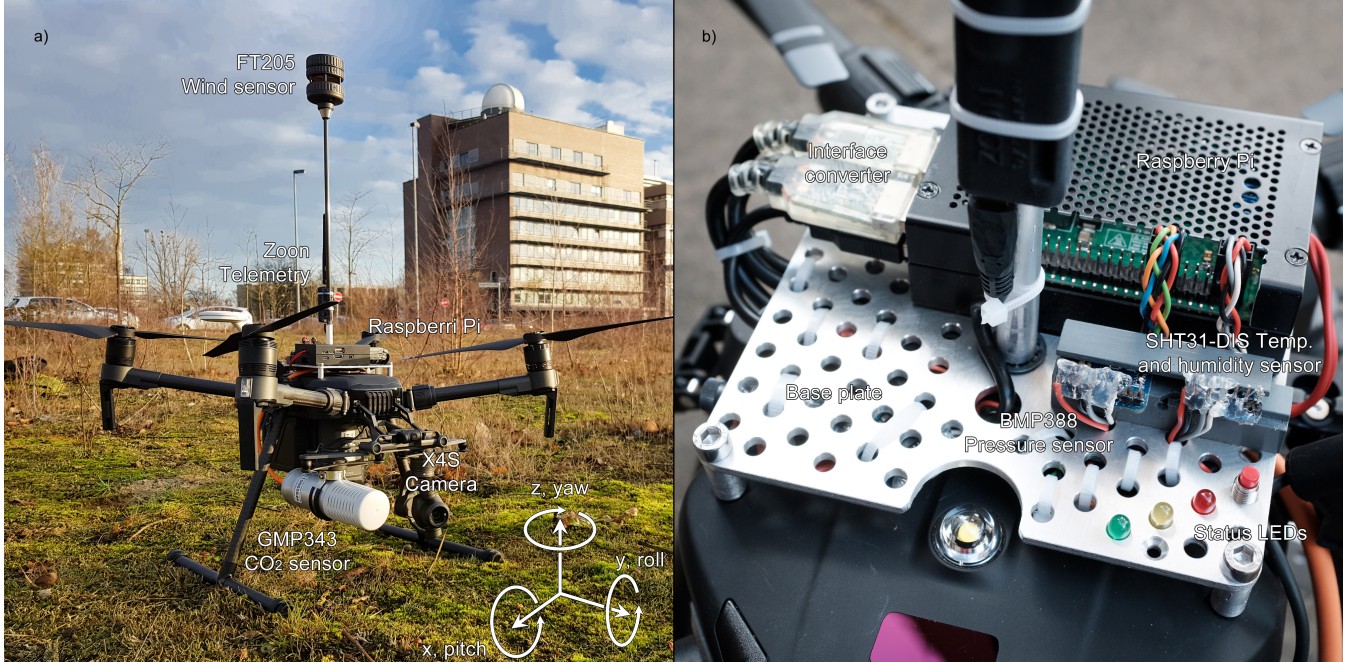

**Figure 1.** UAV and instrumentation. **Left**: Overview including UAV coordinate system as defined by us. **Right**: Zoom on base plate.

The entire payload weights about $1.2\,\text{kg}$ and consumes $6\,\text{W}$ reducing the maximum flight time of 34 minutes to about 24.7 minutes. The reduction in flight time results mainly from the weight of the payload and only marginally from its power consumption. The individual contributions are listed in Tab. 1. All components of the sUAS have a total retail price of about 15k€ including the UAV, which has the highest unit price followed by the $CO_2$ sensor.

## 3 $CO_2$ sensor characterization

Flux quantification of an individual source requires measurements of the difference between elevated concentrations minus undisturbed background concentrations. This means that a constant bias of the measurements is less critical than drifts during flight. Unfortunately some NDIR gas sensors can experience such drifts, e.g., related to temperature (e.g., van Leeuwen et al., 2013; Shusterman et al., 2016; Kunz et al., 2018; Ouchi et al., 2019). The GMP343 corrects for changes in temperature, pressure, or humidity. For this purpose, it measures the temperature with an internal sensor and accepts user inputs for pressure and humidity. Shusterman et al. (2016) and van Leeuwen et al. (2013) found some residual dependencies of the sensor output to temperature, pressure, and humidity and proposed to derive an empirical correction which can be specific for a sensor unit.

We compare roughly one week of continuous measurements of our $CO_2$ sensor with simultaneous measurements of a highly precise (better than $\pm0.3\,\text{ppm}$ without averaging as done here) ABB LGR-ICOS ultra-portable greenhouse gas analyzer





**Table 1.** UAV and instrumentation: weight, power consumption, and contribution to flight time. The reduction in flight time results from the weight and the power consumption of the payload. $^\sharp$Average power consumption during flight including camera and for the total take-off weight of 6022g. $^\flat$Considering only the weight of the camera. $^\natural$Including meteorological sensors, aluminum housing, and power converter.

| Device | Weight [g] | Power [W] | Flight time [min] |
|---|---|---|---|
| UAV DJI M210v2 | 4800 | 836.1$^\sharp$ | 34.0 |
| Camera DJI X4S | 253 | - | -1.9$^\flat$ |
| $CO_2$ Vaisala GMP343 | 360 | 2.4 | -2.7 |
| Wind FTTech. FT205 | 100 | 0.6 | -0.8 |
| Raspberry Pi$^\natural$ | 143 | 2.8 | -1.1 |
| Telemetry Dronee Zoon | 36 | 0.2 | -0.3 |
| Wiring, converters, etc. | 143 | - | -1.1 |
| Plattform and pole mount | 187 | - | -1.4 |
| Total | 6022 | 842.1 | 24.7 |

(*https://new.abb.com*, last access 02.06.2020) in the laboratory under conditions of varying ambient pressure, temperature, and
humidity (Fig. 2).

Air pressure is measured with the BMP388, temperature with the GMP343 internal, and humidity with the SHT31-DIS sensor. During the measurement period, the $CO_2$ concentration changes by more than 200 ppm, the air pressure varies by about 17 hPa, the relative humidity is in the range between about 35% and 60%, and the temperature changes by approximately 6°C. This means, except for $CO_2$, all parameters vary more than we expect for a typical measurement flight.

Throughout the entire comparison time, the $CO_2$ measurements are always close together with an average difference of -0.33 ppm. We compute the 1h running average of the difference between the GMP343 and the LGR instrument. Deviations from zero can safely be assumed to be not dominated by instrumental noise or short term fluctuations. The standard deviation of the running average amounts to 0.89 ppm and we consider it an estimate for potential systematic $CO_2$ drifts during a measurement flight.

The individual soundings scatter around the running average with a standard deviation of 1.76 ppm. However, one of the first test-flights with very variable $CO_2$ concentrations near a stack revealed that always two GMP343 measurements (sampling rate 0.5 Hz) are paired, i.e., they exhibit similar $CO_2$ values within some ppm and Vaisala's technical support confirmed that truly independent measurements are only possible every 4 seconds. In order to account for this, we multiply the derived standard deviation by $\sqrt{2}$ and get 2.49 ppm as estimate for the noise of the individual soundings. This is an upper limit of
the measurement noise because the multiplication by $\sqrt{2}$ would imply that both measurements within a 4 seconds interval are identical, which is not the case.

In order to analyze the drifts of the 1h running average in more detail, we searched for correlations with temperature ($T$ in °C), pressure ($p$ in hPa), absolute humidity ($h_a$ in $g\,m^{-3}$), and the measured $CO_2$ concentration (*GMP343* in ppm). Similar

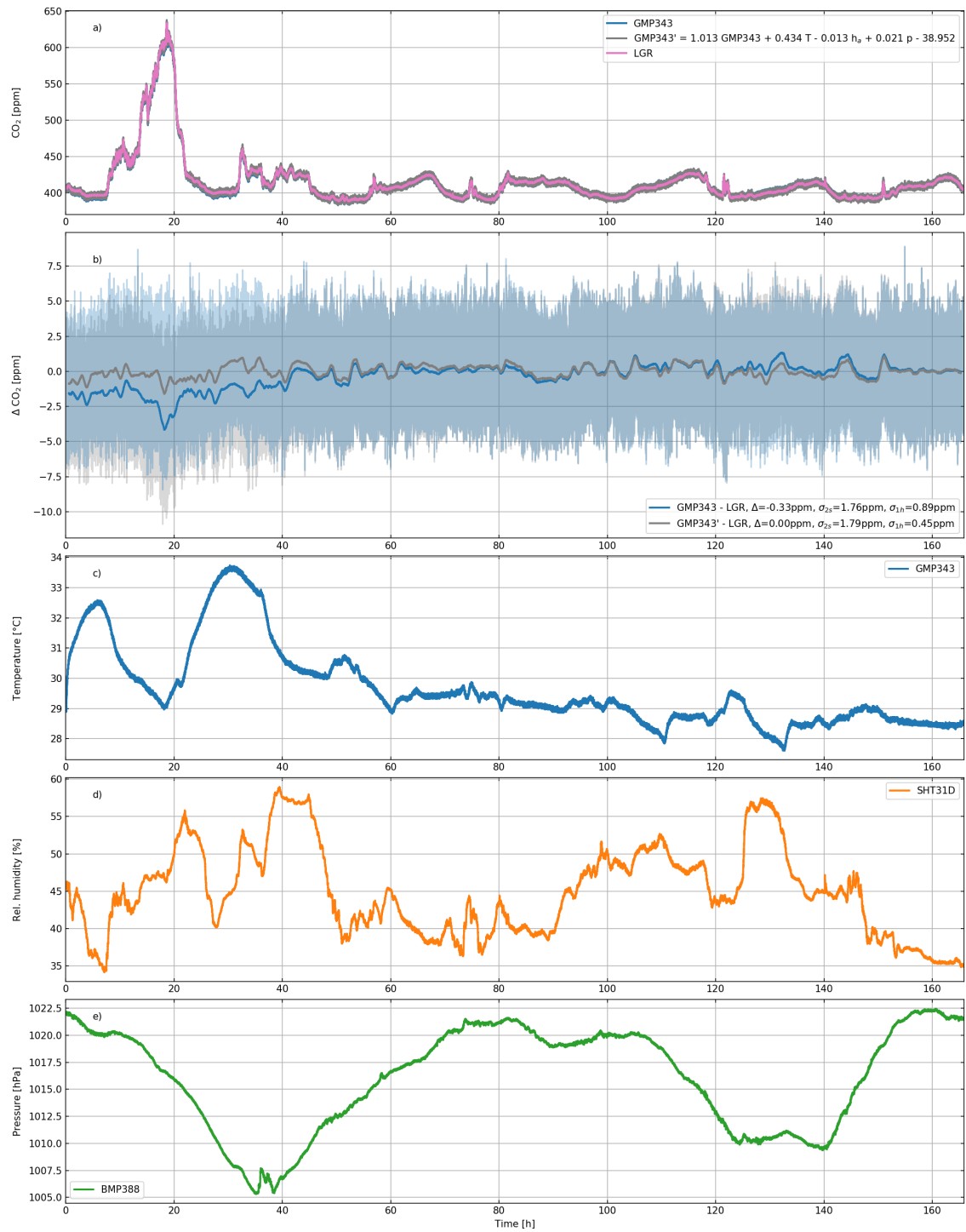

**Figure 2.** Comparison of $CO_2$ concentrations measured with the Vaisala GMP343 $CO_2$ sonde and with a highly accurate ABB LGR-ICOS ultra-portable greenhouse gas analyzer (**a**, **b**) under varying ambient temperature (**c**), relative humidity (**d**), and pressure (**e**) in the laboratory.





to the approach used by van Leeuwen et al. (2013), we fit a linear model with these parameters as explanatory variables and the $CO_2$ concentration of the LGR instrument as response variable and get the following potential correction (in ppm) for our GMP343 sensor:

$$GMP343' = 1.013\,GMP343 + 0.434\,T - 0.013\,h_{\mathrm{a}} + 0.021\,p - 38.952 \tag{1}$$

Note, the absolute humidity is computed from the relative humidity ($h_{\mathrm{r}}$ in %) by:

$$h_{\mathrm{a}} = h_{\mathrm{r}}/100\,SVD \tag{2}$$

wherein $SVD$ is the saturated vapor density (in $\mathrm{g\,m^{-3}}$), which can be well approximated for temperatures between 0°C and 40°C by the following equation (Nave, 2017):

$$SVD \approx 5.018 + 0.32321\,T + 8.1847 \cdot 10^{-3}\,T^2 + 3.1243 \cdot 10^{-4}\,T^3. \tag{3}$$

In principle, Eq. 1 could be used as refinement of the internal corrections of our GMP343 sensor unit and indeed, Fig. 2 suggests that it could reduce potential drifts to 0.45 ppm. However, in terms of expected variability of temperature, pressure, and humidity during a measurement flight, the variability of the correction is small (<1 ppm). The dominating part of the correction comes from the apparent underestimation of 1.3% of the $CO_2$ variability as measured by the GMP343.

However, it can be expected that other factors (especially, wind speed, see Sec. 4) add larger uncertainties to the flux estimates. Therefore, and because more tests would be needed to confirm the robustness of the derived linear model, we decided to use the unmodified, only internally corrected GMP343 measurements as read from the instrument.

## 4 Anemometer calibration

Operating an anemometer near an object that can disturb the local wind field (e.g., a tower) requires calibration to derive the wind speed of the undisturbed wind field. This is particularly the case for the operation aboard an UAV influencing the local wind field not only by its static parts but especially by the downwash of the rotors. We tried to reduce both effects by mounting the anemometer on a 7 mm diameter carbon fiber pole about 36 cm above the rotor plane. A larger distance to the rotor plane would of course reduce the influence of the rotors, but the flight behavior could suffer if the distance becomes too large. In addition, pitch and roll maneuvers of the UAV would result in larger relative velocities of the anemometer.

Let $m'$ be the horizontal wind vector (in x- and y-direction of the UAV coordinate system, see Fig. 1) as measured by the anemometer. We define the calibrated wind vector $u'$ (also in UAV coordinates) by the following calibration function which allows for scaling and rotating the original measurements by matrix $\mathbf{A}$ and for adding a constant vector $b$.

$$u' = \mathbf{A}\,m' + b \tag{4}$$

Applying the rotation matrix $\mathbf{R}$ with $\gamma$ being the azimuthal orientation (yaw) of the UAV

$$\mathbf{R} = \begin{pmatrix} \cos\gamma & -\sin\gamma \\ \sin\gamma & \cos\gamma \end{pmatrix} \tag{5}$$



results in the calibrated wind at the anemometer but in geographic coordinates:

$$u = \mathbf{R}^{-1} \left( \mathbf{A} \, m' + b \right). \tag{6}$$

The wind at the anemometer is a superposition of the wind relative to ground $w$ and the headwind because of the velocity $v$ of the UAV ($u = w - v$), so that the wind relative to ground becomes

$$w = \mathbf{R}^{-1} \left( \mathbf{A} \, m' + b \right) + v. \tag{7}$$

A straight forward approach to derive the coefficients of $\mathbf{A}$ and $b$ would be to make measurements in a wind tunnel at different wind speeds $w$ and orientations $\gamma$ of the UAV while continuously hovering at the same position ($v = 0$). However,

this procedure is impractical because it requires a relatively large wind tunnel and hovering in place becomes more difficult if GPS is not available.

Consequently we selected a different calibration strategy: under the assumption of a stationary wind field, we performed a calibration flight with a flight pattern that enables $w$ and the coefficients of $\mathbf{A}$ and $b$ to be derived simultaneously. The flight pattern has to include back and forth movements of the UAV in two ideally perpendicular directions and has to be repeated

with varying velocities and orientations (see Fig. 3).

Specifically, we solve Eq. 7 for the uncalibrated anemometer measurement

$$m' = \mathbf{A}^{-1} \left[ \, \mathbf{R} \, (w - v) - b \, \right] \tag{8}$$

and perform a least squares fit to find the coefficients of $\mathbf{A}$, $b$, and $w$. In order to best fulfill the stationarity assumption of the wind field, we performed the calibration flight on a day with relatively calm conditions and low gusts. Additionally, we use

only measurements with a horizontal total acceleration of less than $0.5 \, \mathrm{m \, s^{-2}}$. The results for the free fit parameters

$$\mathbf{A} = \begin{pmatrix} 0.881 \pm 0.010 & 0.018 \pm 0.010 \\ -0.021 \pm 0.020 & 0.832 \pm 0.010 \end{pmatrix}, \; b = \begin{pmatrix} 0.074 \pm 0.040 \\ -0.058 \pm 0.038 \end{pmatrix}, \; v = \begin{pmatrix} 1.064 \pm 0.041 \\ 1.063 \pm 0.041 \end{pmatrix} \tag{9}$$

show that the calibration scales the measured wind speed in x-direction with 0.881 and 0.832 in y-direction. A rotation is basically not applied and there is only a very small constant offset correction below $0.1 \, \mathrm{m \, s^{-1}}$ needed.

As shown in Fig. 4 (a, b), the fitted wind measurements (right side of Eq. 8, green) agree well with the actually measured

values (left side of Eq. 8, blue). The derived wind relative to ground $w$ (Fig. 4 c, d, green) scatters around the mean values of $1.06 \, \mathrm{m \, s^{-1}}$ in east and west direction. We repeated the calibration experiment and obtained similar coefficients even though the conditions were less ideal with a slightly larger average wind speed of $2.40 \, \mathrm{m \, s^{-1}}$ and more gusts (not shown).

We estimate the scatter of the wind components in two different ways. i) We compute the standard deviation of the wind components and consider it an upper limit of the scatter as it includes not only the noise of the measurements (including

calibration) but also the actual variability of the wind. ii) Under the assumption that the actual wind varies only smoothly, we estimate the noise of the measurements by the standard deviation of the difference of successive measurements divided by $\sqrt{2}$. As just mentioned, the first method (east component: $0.55 \, \mathrm{m \, s^{-1}}$, north component: $0.66 \, \mathrm{m \, s^{-1}}$) overestimates the true noise.





**Figure 3.** Flight pattern of the anemometer calibration experiment. From top to bottom: location (**a**), distance to the (arbitrarily chosen) origin in east direction (**b**), distance to the (arbitrarily chosen) origin in north direction (**c**), ground speed (**d**), and yaw (**e**) of the UAV. Valid measurements are shown in dark blue, filtered measurements in light blue. The filtering removes measurements that lie not within the calibration pattern or with UAV accelerations larger than $0.5\,\mathrm{m\,s^{-2}}$.





**Figure 4.** Results of the anemometer calibration experiment. **a)** X-component of the uncalibrated wind measurement and roll (blue and red, left side of Eq. 8) as well as corresponding fits (green and orange, right side of Eq. 8). **b)** Y-component of the uncalibrated wind measurement and pitch as well as corresponding fits. **c)** Wind speed relative to ground in east direction for the calibrated (green) and uncalibrated (blue) anemometer as well as computed from the attitude of the UAV (orange). **d)** Wind speed relative to ground in north direction. Histograms of wind speed and direction for the calibrated (**e**) and uncalibrated (**f**) anemometer as well as computed from attitude (**g**).





However, the second estimate (both components: $0.33\,\mathrm{m\,s^{-1}}$) is probably too optimistic because it assumes that slow changes of the wind only come from the actual variability of the wind but not from the instrument or the calibration method.

Comparing Fig. 4 f) and g) shows that the calibration of the anemometer narrows the histogram of the derived wind relative to ground. The average calibrated total wind speed amounts to $1.5\,\mathrm{m\,s^{-1}}$. The scatter of the total wind speed has been computed in the same manner as done for the components and amounts to $0.29\,\mathrm{m\,s^{-1}}$ and $0.60\,\mathrm{m\,s^{-1}}$ for the optimistic and pessimistic computation, respectively. Using the uncalibrated measurements would result in a higher average total wind speed of $1.77\,\mathrm{m\,s^{-1}}$ and a reduced precision between $0.40\,\mathrm{m\,s^{-1}}$ and $0.86\,\mathrm{m\,s^{-1}}$.

For stationary conditions, pitch and roll could serve as proxies for the wind speed in x- and y-direction because the UAV has to tilt in order to resist the wind. We use the same calibration method to find calibration coefficients for pitch and roll as explanatory variables instead of the measured uncalibrated anemometer signal. Overall this relatively simple method to get wind information from the attitude of the UAV instead of an anemometer gives also reasonable results but with the drawbacks of a lower precision (between $0.74\,\mathrm{m\,s^{-1}}$ and $0.89\,\mathrm{m\,s^{-1}}$) and that a stricter filtering for non-static conditions has to be applied

because of pitch and roll due to acceleration. Specifically, this was the driver to use only measurements with a horizontal total acceleration of less than $0.5\,\mathrm{m\,s^{-2}}$. Choosing considerably larger threshold values results in much larger spikes in the derived attitude-based wind components.

Note that the anemometer-based wind measurements are also somewhat influenced by accelerations of the UAV but to a far lesser extent. This comes from pitch and roll maneuvers of the UAV resulting in movements of the anemometer at the top of

the pole and from variations of the rotor downwash.

## 5   Validation using ICOS measurements

In order to assure that all instruments of the sUAS work as expected in-flight, we performed two validation flights on 09.04.2020 at the ICOS atmospheric station Steinkimmen (STE) near Bremen, Germany (Fig. 5) which is hosted on a 285 m high broadcasting tower of the NDR (Norddeutscher Rundfunk). During the flights the average wind speed was $2.79\,\mathrm{m\,s^{-1}}$ mainly in east

direction (Fig. 5, c).

At this ICOS station, greenhouse gas concentrations ($CO_2$, $CH_4$, and $N_2O$) and meteorological parameters (e.g., wind speed and direction, temperature, and humidity) are continuously measured at five different levels with altitudes of 32 m, 82 m, 127 m, 187 m, and 252 m (*https://icos-atc.lsce.ipsl.fr/STE*). We use ICOS Level 1 data with a temporal resolution of one minute (Kubistin et al., 2020).

The $CO_2$ concentration is measured with a Picarro G2301 cavity ring-down spectroscopy analyzer and each measurement level is sampled for five minutes so that a full measurement cycle takes 25 min. The one minute averaged $CO_2$ concentrations are precise to about $\pm 0.1$ ppm ($1\sigma$). The wind in each height is measured with Thies Clima ultrasonic 2D anemometers (accuracy: $0.1\,\mathrm{m\,s^{-1}}$ for wind speeds up to $5\,\mathrm{m\,s^{-1}}$) and is also averaged to one minute intervals. Temperature and humidity in each height are measured with Vaisala HMP155 sensors.



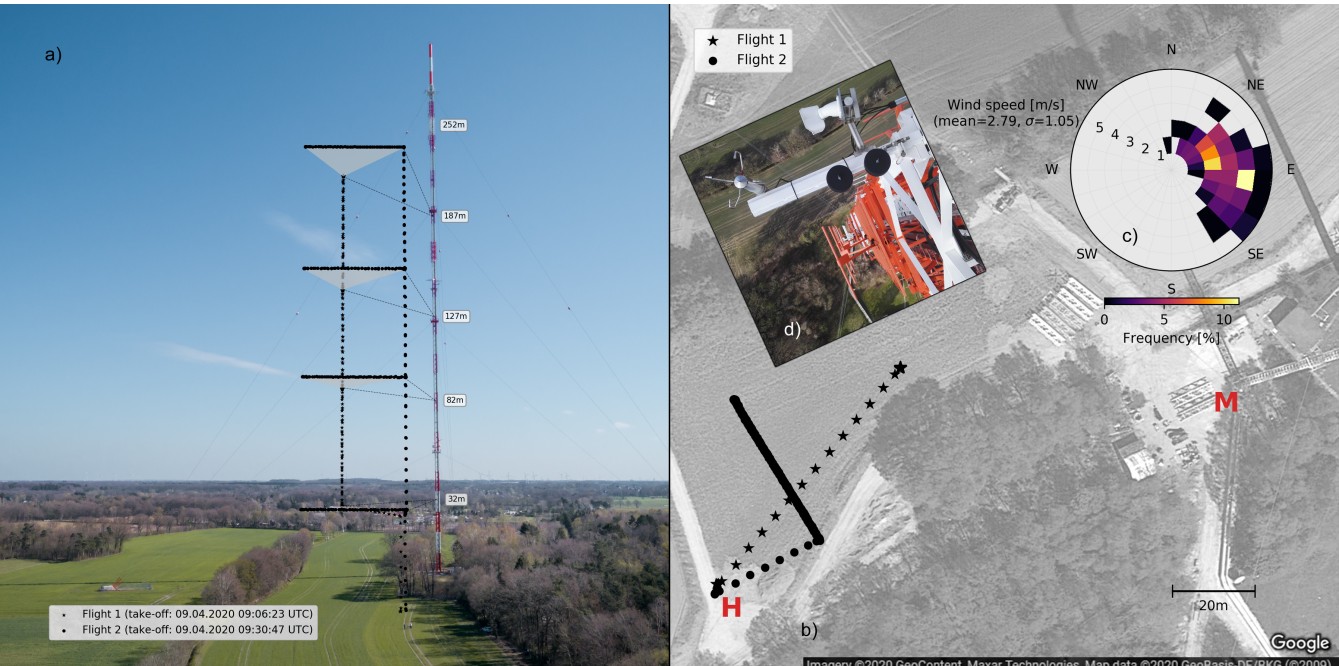

**Figure 5.** Overview of the validation flights on 09.04.2020 at the ICOS atmospheric station Steinkimmen (STE) near Bremen hosted on a broadcasting tower of the NDR. **a)** Positions of measurements of both flights projected onto an aerial photograph. **b)** Top view of the measurement site including launch point (H), position of measurements (stars and circles), and position of the lattice tower (M). **c)** Histogram of wind speed and direction. **d)** ICOS instruments and orientation of the lattice tower (image courtesy of the German meteorological service DWD).

We performed two validation flights with a distance of about 100 m to the tower. In order not to risk signal interference (99% or 700 kW of the broadcasting power is emitted above 200 m), we only visited the four lower measurement levels.

During the first flight, the UAV twice ascended to 187 m and descended level by level to 32 m, hovering about 45 s at each measurement level. During the second flight we visited each measurement level only once, but instead of hovering, we flew three times back and forth on a 40 m long track so that the UAV was always moving while measuring. In total there are 12
measurement sequences (Fig. A1 and 6, S1 – S12) where the UAV visited one of the four measurement levels. The flight pattern is shown in detail in Fig. A1.

For each measurement level, we linearly interpolate the ICOS $CO_2$, wind, temperature, and humidity to the times of our measurements. We only use those measurements for comparisons which fall into one of the 12 measurement sequences and for which the UAV acceleration is below $0.1 \, \text{m} \, \text{s}^{-2}$. We here use a rather strict threshold for the maximum accepted acceleration
because we wanted the results to be dominated by steady-state conditions, as expected when flying at a constant speed.

Our $CO_2$ measurements (Fig. 6, a) have basically no systematic offset relative to the ICOS observations (0.07 ppm) and the standard deviation of the difference amounts 2.12 ppm. The ICOS $CO_2$ measurements vary negligibly during both flights and

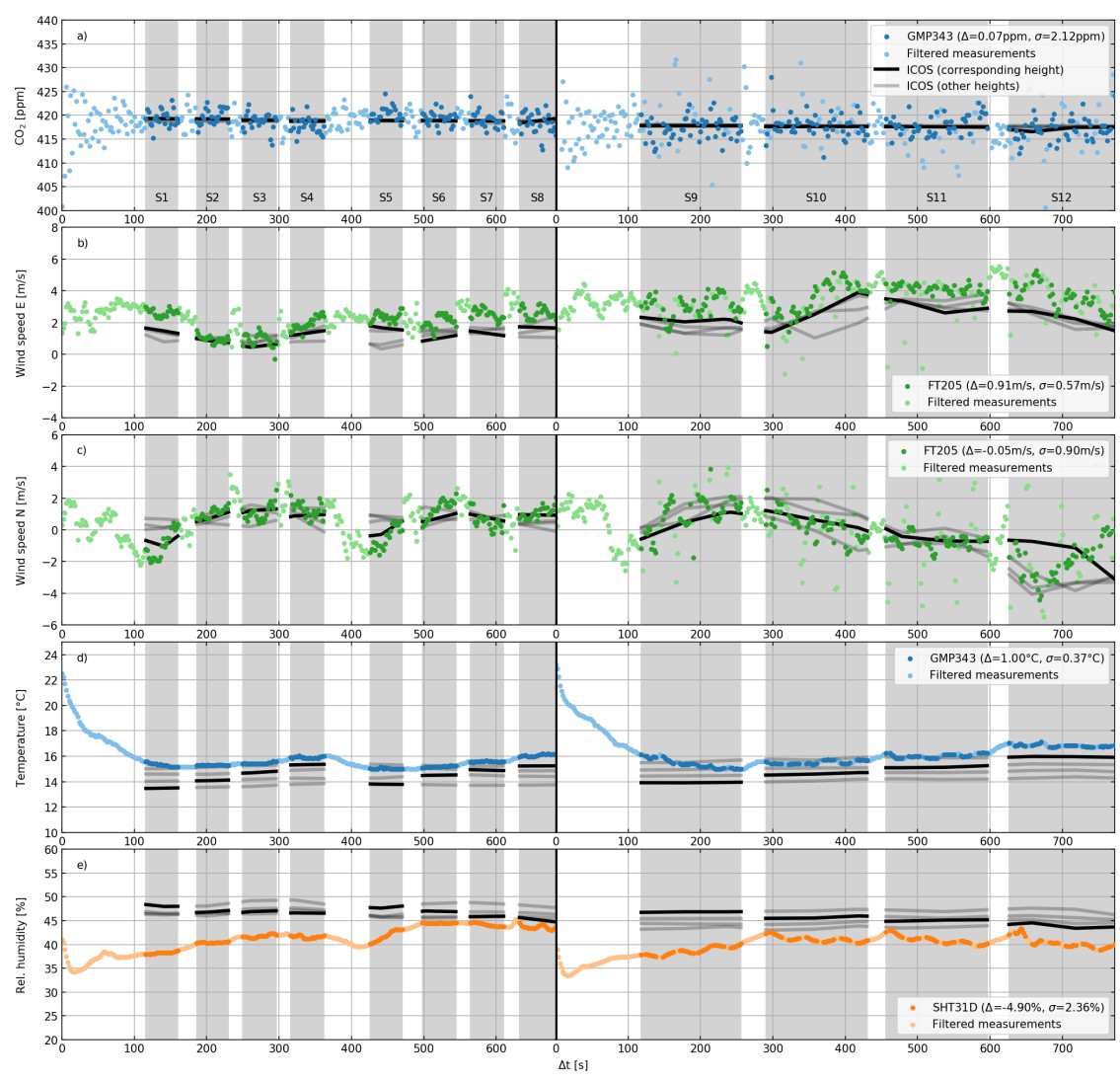

**Figure 6.** Measurements and results from the validation flights at the ICOS atmospheric station Steinkimmen (STE), Germany, 09.04.2020. From top to bottom: $CO_2$ mole fraction (**a**), wind speed in east direction (**b**), wind speed in north direction (**c**), temperature (**d**), relative humidity (**e**). **Left**: flight 1. **Right**: flight 2. Gray areas correspond to measurement sequences S1 – S12 during which the UAV visited one of the ICOS measurement levels. Light colors represent filtered measurement with an UAV acceleration larger than $0.1\,\mathrm{m\,s^{-2}}$. Bold colors represent measurements used for the validation.





from height to height (Fig. 6, a) which indicates well mixed atmospheric conditions so that no significant representation errors, i.e., differences due to atmospheric variability, have to be expected. The measured $CO_2$ scatters slightly more in the second

flight compared to the first (especially when the filtered measurements are included). This could be due to fluctuating pressure conditions caused by the rotors during the accelerations when flying back and forth in sequence S9 – S12. As discussed in Sec. 3, we multiply the derived scatter with $\sqrt{2}$ and get 3.00 ppm as estimate for the in-flight precision of our $CO_2$ measurements. This is similar to the noise error estimated in the laboratory (Sec. 3) and agrees with the precision as specified by the manufacturer.

We find no significant drifts from height to height which would hint, e.g., at strong uncorrected pressure or temperature dependencies and we also find no significant systematic offsets between both flights. According to Sec. 3, we estimate that $CO_2$ enhancements can be measured in-flight with an accuracy (or trueness) of 1.3% or 0.9 ppm, whichever is larger.

The average of the north component of the wind speed (Fig. 6, c) is only slightly larger ($0.1\,\mathrm{m\,s^{-1}}$) compared to the ICOS measurements and the standard deviation of the difference amounts to $0.90\,\mathrm{m\,s^{-1}}$. The relative large scatter of the difference to

ICOS is mainly driven by a poor agreement in S12. As visible in Fig. 5 (a and b), the tower is located close to a small piece of forest while the flight track is above a relatively free field which can lead to significant differences of the measurements at 32 m height. Additionally, it shall be noted that the second flight has a larger distance to the tower compared to the first flight. When considering only S1–S11, the average difference becomes $0.02\,\mathrm{m\,s^{-1}}$ and the standard deviation of the difference reduces to $0.67\,\mathrm{m\,s^{-1}}$.

The east component of the wind (Fig. 6, b) is on average $0.9\,\mathrm{m\,s^{-1}}$ larger compared to the ICOS measurements and the standard deviation of the difference is $0.57\,\mathrm{m\,s^{-1}}$. The ICOS anemometers are mounted on short outriggers at the west side of the tower, leading to a possible underestimation of the ICOS measurements especially in east direction due blockage of eastbound winds as prevailing during the validation flights (Fig. 5, b and c). This would be consistent with the generally larger east component of the wind speed measured with the sUAS compared to the ICOS measurements, independent of the

orientation of the sUAS so that a systematic overestimation in one direction in UAV coordinates can be excluded as explanation.

The average of the scatter of the north and east component is $0.62\,\mathrm{m\,s^{-1}}$ (excluding S12 of the north component). In addition to the noise of our measurements, this includes the representation error and the noise of the ICOS measurements. Therefore, it can be considered an upper limit of the scatter of our measurements of a horizontal wind component. This agrees well with the upper limit of the scatter determined in Sec. 4 and compares to the lower limit of $0.33\,\mathrm{m\,s^{-1}}$ computed in the same section. For

convenience, we estimate that the precision of the individual measurements of a horizontal wind component lies in the middle of both values ($0.48\,\mathrm{m\,s^{-1}}$).

In order to estimate the systematic uncertainties of our wind component measurements, we compute the average differences in sequence S1–S11 of the north component. The standard deviation of these values amounts to $0.34\,\mathrm{m\,s^{-1}}$ and is considered an estimate of the accuracy of our wind component measurements.

Due to the overestimation of the east component (or underestimation by ICOS), also the total horizontal wind speed is larger than the ICOS observations. On average, the difference amounts to $0.84\,\mathrm{m\,s^{-1}}$ and has a standard deviation of $0.51\,\mathrm{m\,s^{-1}}$ (when excluding S12). We estimate from the latter value and from the lower limit of the scatter (Sec. 4) that the precision



of the individual measurements of the total horizontal wind speed is $0.40\,\mathrm{m\,s^{-1}}$, which is consistent with the manufacturer's specification of $\pm0.3\,\mathrm{m\,s^{-1}}$ for wind speeds below $16\,\mathrm{m\,s^{-1}}$. The accuracy of the total horizontal wind speed is $0.33\,\mathrm{m\,s^{-1}}$ and
has been estimated as for the wind components.

The temperature measured with the internal temperature sensor of the GMP343 $CO_2$ sonde is typically about 1 K larger compared to the ICOS measurements (Fig. 6, d). However, it can reasonably well reproduce the temperature profile with smaller values at higher altitudes, even though, especially within the first 3–4 minutes after launch (S1, S2, and S9) one can clearly see the hysteresis of the sensor resulting in an overestimation of up to 2 K. A potential explanation of the general
overestimation could be the heating of the GMP343 $CO_2$ sonde, which is intended to reduce the possibility of condensation on the optical components but which could also slightly warm the temperature sensor. Relative humidity (Fig. 6, e) is typically up to 10% lower than measured by ICOS and the sensor appears to have a relatively long response time in the order of some minutes.

## 6   Elevated $CO_2$ concentrations downwind of an industrial facility

In order to demonstrate how the sUAS can be used to measure elevated $CO_2$ concentrations of a nearby source, we performed two flights on 10.04.2020 near the ExxonMobil natural gas processing facility in Großenkneten about 40 km east of Bremen, Germany (Fig. 7). In this facility, hydrogen sulfide ($H_2S$) is removed from natural gas (i.e., sour gas is "sweetened"), which is an energy intense process. According to the European Pollutant Release and Transfer Register (E-PRTR), the annual $CO_2$ emissions of the facility declined from 1260 kt in 2014 to 1010 kt in 2017 which is the most recent value in the E-PRTR
data base (*https://prtr.eea.europa.eu*, last access 11.06.2020). Similar values for the $CO_2$ equivalent emissions can be found in the most recent (2019) list of stationary facilities in Germany that are subject to emissions trading published by the German emissions trading authority (DEHSt, *https://www.dehst.de*, last access 11.06.2020) of the German Environment Agency, but in addition to the E-PRTR data base, this list also includes values for 2018 (855 kt) and 2019 (892 kt).

It can be expected that most of the $CO_2$ emissions are released by the two 150 m high stacks, prominently visible in Fig. 7
(S1). In 2014 a combined heat and power plant was put into operation and from information on the ExxonMobil website (*https://www.erdgas-aus-deutschland.de*, last access 11.05.2020), one can roughly estimate that about 17% of the total emissions may be emitted via its 34 m high stack (Fig. 7, S2). Moreover, an unknown but likely small proportion is emitted by gas flaring (Fig. 7, S3). We roughly estimate that 740 kt $CO_2$ have been released via the main stacks in 2019. For later emissions or the inner-annual variability, we do not have an estimate.

Targeting at the emissions of the main stacks, we flew two vertical cross-sections roughly perpendicular to the average wind direction with a length of 272 m and altitudes ranging from 120 m to 200 m. The second cross-section has been shifted by roughly 70 m westwards relative to the first one in order to get a denser sampling in the region with the largest expected enhancements. Number of tracks and velocity of the UAV relative to ground have been chosen to realize a similar sampling density in horizontal (9.1 m) and vertical (8.9 m) direction for 2 s measurement intervals. Due to flight regulations, we limited



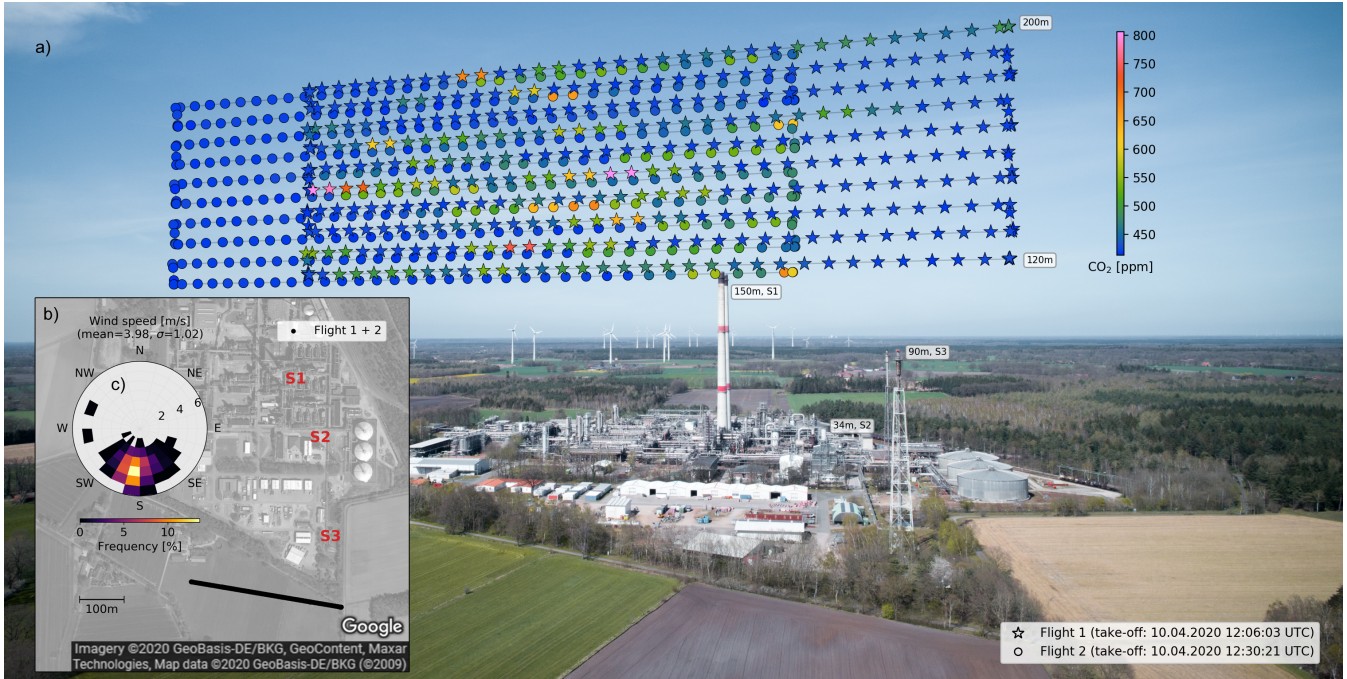

**Figure 7.** Overview of two flights downwind of the ExxonMobil natural gas processing facility in Großenkneten 40 km east of Bremen, Germany, 10.04.2020. **a)** $CO_2$ measurements of both flights projected onto an aerial photograph taken from a height of 80 m. Note: only for the purpose of plotting, we added 3 m to the altitudes of the first flight. **b)** Top view of the measurement site including positions of the sources S1 – S3, flight track, and histogram of wind speed and direction (**c**).

the maximum altitude to 200 m, although we anticipated that parts of the plume would probably have risen above that. The details of the flight pattern are shown in Fig. A2.

The $CO_2$ mass flux density perpendicular through the cross-section resulting from the facility's emissions can be computed by:

$$F = w_\perp \, \rho_{\mathrm{air}} \, \Delta CO_2 \, M_{\mathrm{CO_2}}. \tag{10}$$

Here $w_\perp$ is the wind speed normal (perpendicular to the cross-section), $\rho_{\mathrm{air}}$ is the molar density of air, $\Delta CO_2$ is the $CO_2$ enhancement caused by the facility's emissions, and $M_{\mathrm{CO_2}}$ is the molar mass of $CO_2$. Strictly speaking, Eq. 10 represents only the mass flux density due to advection and we assume that diffusion can be neglected, which is the case for wind speed normals larger than about $2 \, \mathrm{m \, s^{-1}}$ (Sharan et al., 1996). The sUAS measures all quantities needed to compute the $CO_2$ flux density for each measurement interval and Fig. 8 shows the quantities needed to compute Eq. 10.

In order to derive $\Delta CO_2$ from the measured $CO_2$ mole fraction, we manually define measurement intervals which we consider to represent background concentrations not disturbed by the emissions of the facility. For each of these intervals (gray background in Fig. 8, a), we compute the average $CO_2$ mole fraction and estimate the background concentration for





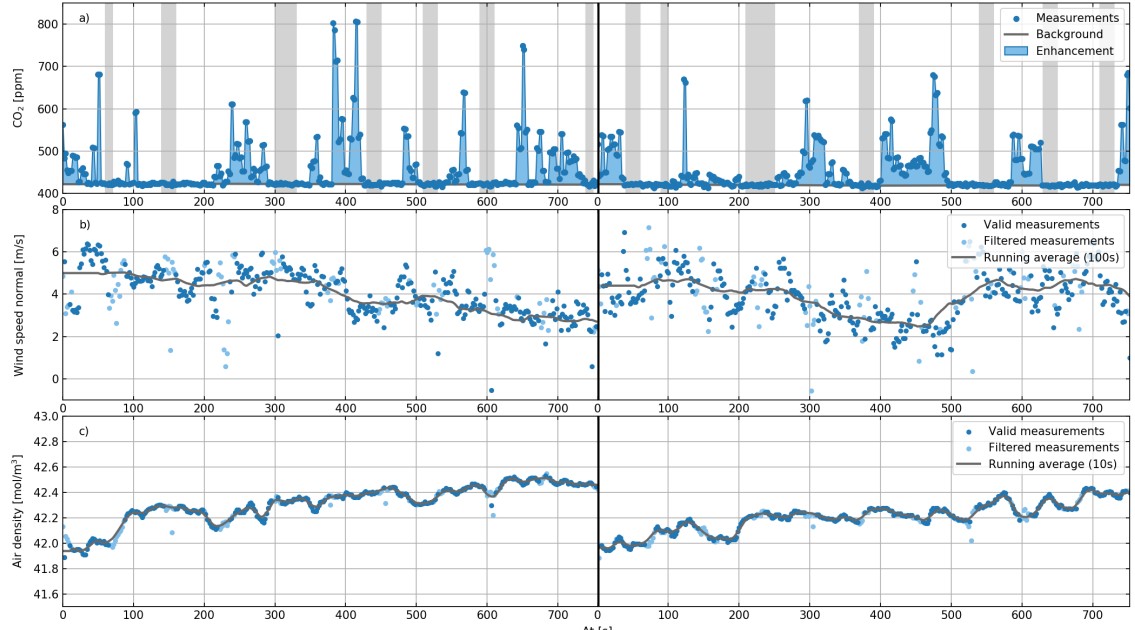

**Figure 8.** Measurements and results from two flights downwind of the ExxonMobil natural gas processing facility in Großenkneten 40 km east of Bremen, Germany, 10.04.2020. **a)** $CO_2$ mole fraction as well as background values (gray line) estimated from measurements within light gray areas and $CO_2$ enhancement (light blue area). **b)** Wind speed normal including 100 s running average (gray line), filtered measurements with UAV accelerations larger than $0.1\,\mathrm{m\,s^{-2}}$ (light blue), and valid measurements (bold blue) used to compute the running average. **c)** Air density including a 10 s running average and filtering information (light and bold blue) as for the wind speed normal. **Left**: flight 1. **Right**: flight 2.

each sounding by linear interpolation. Under the assumption of no additional nearby upwind sources, we compute the $CO_2$ enhancement caused by the facility's emissions by the difference between the actually measured and the estimated background

concentration (light blue area in Fig. 8, a). The $CO_2$ enhancement is often in the order of 100 ppm and reaches maximum values of almost 400 ppm.

For each sounding, we compute the wind speed normal from the calibrated wind measurements and filter out wind measurements with UAV accelerations larger than $0.1\,\mathrm{m\,s^{-2}}$. The resulting gaps are filled with linearly interpolated values and in order to reduce the noise of the wind measurements, we compute a 100 s running average (Fig. 8, b). The wind speed reduces with

height superimposed by a refreshing of the wind towards the end of the second flight, which we also noticed on ground. The average wind speed normal amounts to $4.0\,\mathrm{m\,s^{-1}}$.

By applying the ideal gas law, we compute the molar air density, which is shown in Fig. 8 (c). We use the same filtering and interpolation as for the wind measurements, but because of less relative noise, we compute a running average with a smoothing kernel of only 10 s instead of 100 s. The most noticeable feature visible in the air density plot (Fig. 8, c) are the increasing

values for lower heights due to the larger pressure.





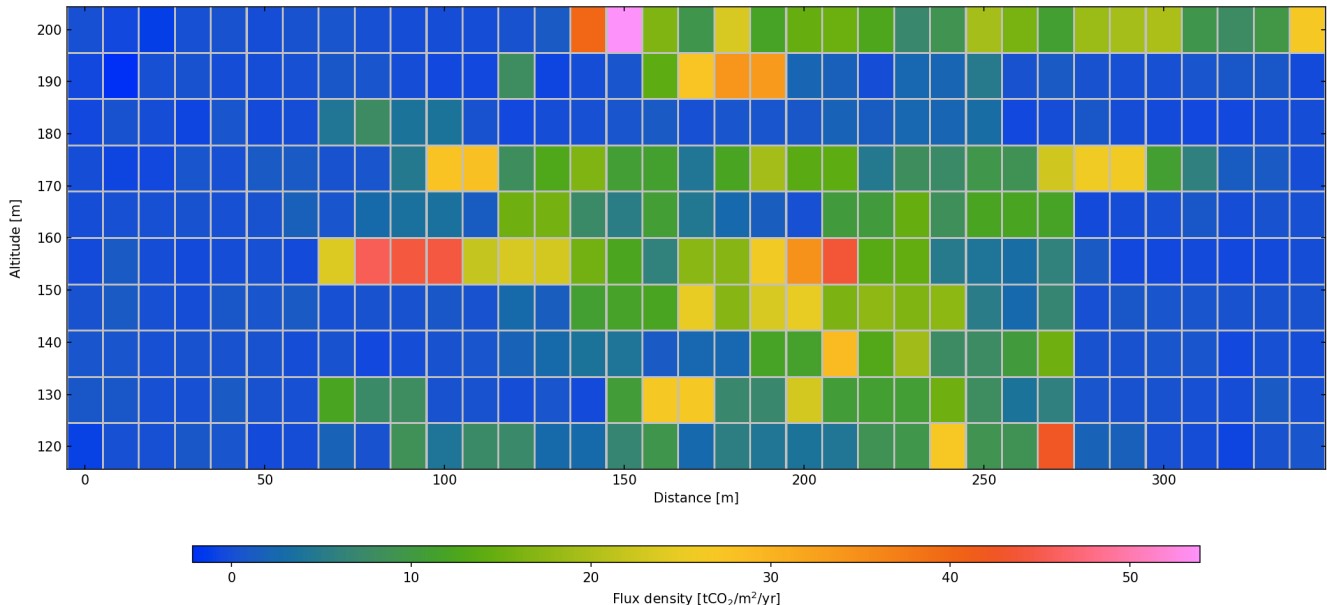

**Figure 9.** $CO_2$ mass flux density perpendicular to the plume cross-section computed from the measurements of two flights downwind of the ExxonMobil natural gas processing facility in Großenkneten 40 km east of Bremen, Germany, 10.04.2020.

We compute the flux density according to Eq. 10 and grid it with a spacing of 10 m horizontally and 8.89 m vertically ensuring that each grid box includes at least one data point (Fig. 9).

Integrating the flux density field results in a total flux of 196 $ktCO_2$ $yr^{-1}$. In order to estimate the uncertainty of this value due to instrumental noise, we compute the total flux from measurements to which we add Gaussian noise with a standard deviation according to the instrument characteristics and the results of the ICOS validation (Sec. 5: 3 ppm for $CO_2$, 0.48 m s$^{-1}$ for the wind speed normal component, 1°C for temperature, and 1 hPa for pressure). We repeat this 100 times and get a standard deviation of 2.5 $ktCO_2$ $yr^{-1}$ which corresponds to the stochastic $1\sigma$ uncertainty.

We also estimate the influence of potential systematic instrumental errors summarized in Sec. 5. The systematic $CO_2$ uncertainty (1.3%) corresponds to a systematic total flux uncertainty of 2.5 $ktCO_2$ $yr^{-1}$. The systematic wind component uncertainty (0.34 m s$^{-1}$ = 8.3%) translates to a flux uncertainty of 16.3 $ktCO_2$ $yr^{-1}$. Assuming 1°C to be the systematic uncertainties of the temperature measurement results in total flux uncertainties of 0.7 $ktCO_2$ $yr^{-1}$ (3.5‰). Systematic uncertainties introduced by the pressure measurements can safely be neglected.

Adding up the variances of the systematic uncertainties plus the stochastic uncertainty results in a total uncertainty of 16.7 $ktCO_2$ $yr^{-1}$ or 8.5%. It shall be noted that this uncertainty estimate only includes systematic and random uncertainties introduced by the instruments. There are other effects which also can result in significant differences between the inferred total flux and the facility's emissions such as turbulence (i.e, short term fluctuations due to non-steady meteorology) or undersampling of the plume morphology. Most of such effects will average out with increasing number of flights or increasing flight



time. How well this works and how large the corresponding uncertainties actually are can be assessed, e.g., by analyzing the variability of results from multiple successive flights or by high resolution plume simulations. However, such assessments are
beyond the scope of this publication.

The inferred total flux of $196 \pm 17 \, \text{ktCO}_2 \, \text{yr}^{-1}$ is significantly lower than our rough estimate of the annual emissions of the main stacks for the year 2019 based on DEHSt ($740 \, \text{ktCO}_2 \, \text{yr}^{-1}$). The uppermost tracks of both flights contain some significantly elevated $CO_2$ measurements (Fig. 7, 8, 9) strongly indicating that we have only seen parts of the plume and we assume that most of the plume has risen above 200 m. In addition, we see some non-background values in the lowermost tracks
which could either come from the nearby gas flare but also indicate that parts of the plume may have been below 120 m.

Note also, the fact that we use $\text{ktCO}_2 \, \text{yr}^{-1}$ as unit for the total flux does not imply that we actually determine annual emissions; our measurements still only correspond to a snapshot of the current situation and the unit has only been chosen because it is commonly used. The estimation of annual emissions usually requires repeated flights throughout the year and assumptions on the variability of the actual emissions have to be made to estimate the corresponding uncertainties (Velazco
et al., 2011).

## 7   Summary and Conclusion

We introduced a small sUAS measuring all atmospheric quantities needed to quantify the $CO_2$ emissions of a nearby point source from its downwind mass flux. The entire sUAS weights about 6 kg and has been build from commercially available components, which allowed us to realize an affordable but reliable system in a relatively short development phase. In order
to quantify the atmospheric $CO_2$ mass flux without the need for any ancillary data, the payload includes a $CO_2$ probe, an anemometer, and sensors for temperature, pressure, and relative humidity. The onboard computer uses a separate radio data link, which allows the flight pattern to be adapted to the $CO_2$ and wind conditions in-flight.

With special emphasis on $CO_2$ and wind, we performed calibration, comparison, and validation experiments in the laboratory and in-flight. We introduced a method to calibrate the anemometer under flight conditions when the headwind has to be
accounted for and rotor downwash can influence the local wind field. We validated our $CO_2$ and wind measurements by comparison with ICOS measurements at the 285 m high NDR broadcasting tower in Steinkimmen near Bremen, Germany.

According to the validation experiment, an upper limit for the $1\sigma$ single sounding precision of our $CO_2$ measurements during a flight is 3 ppm (measurement interval = 2 s) and from a linear analysis of correlated errors in the laboratory, we conclude that $CO_2$ enhancements can be measured with an accuracy of 1.3% or 0.9 ppm, whichever is larger.
Our anemometer calibration method derives the free fit parameters of a linear calibration model accounting for scaling, rotation, and a potential constant bias. For this purpose it analyzes wind measurements taken while following a suitable flight pattern and assuming stationary wind conditions. From the calibration and validation experiments, we estimated that we can retrieve the total horizontal wind speed relative to ground during a flight with a $1\sigma$ single measurement precision of $0.40 \, \text{m s}^{-1}$ and an accuracy of $0.33 \, \text{m s}^{-1}$. The precision is similar to the root mean square errors of $0.5 \, \text{m s}^{-1}$ and $0.4 \, \text{m s}^{-1}$ found by





Palomaki et al. (2017) and Shimura et al. (2018), respectively, but in both cases a bias of $0.5\,\mathrm{m\,s^{-1}}$ has previously been subtracted.

For the purpose of flux estimation, the uncertainty of the wind speed components is more relevant than the uncertainty of the total wind speed. We estimated that these can be retrieved with a $1\sigma$ single measurement precision of $0.48\,\mathrm{m\,s^{-1}}$ and an accuracy of $0.34\,\mathrm{m\,s^{-1}}$. This compares to an accuracy of $0.5\,\mathrm{m\,s^{-1}}$ of the wind components measured aboard a manned aircraft

by Krings et al. (2018).

Additionally, we showed that our calibration method can also be used to derive wind information from the attitude of the UAV. The inferred wind speeds are reasonably consistent with those of the calibrated anemometer but feature a reduced quality. It cannot be excluded that the attitude base wind retrieval might be significantly improved in the future, but for the time being, our findings are consistent with those of Palomaki et al. (2017) and Barbieri et al. (2019) who concluded that sonic anemometers

provide the most accurate wind information from multi-rotor platforms.

During two flights, we measured the $CO_2$ enhancement downwind of the ExxonMobil natural gas processing facility in Großenkneten about 40 km east of Bremen, Germany and demonstrated how the measurements of the sUAS can be used to infer mass fluxes of atmospheric $CO_2$ related to the emissions of the facility. We aimed at the emission of the two 150 m high main stacks of the facility and centered the cross-sectional flight patterns according to the wind direction and found

enhancements of up to almost 400 ppm. Integration over the inferred flux densities resulted in a total cross-sectional flux of $196\pm17\,\mathrm{ktCO_2\,yr^{-1}}$. This is significantly lower than our rough estimate of $740\,\mathrm{ktCO_2\,yr^{-1}}$ for the annual emissions of the main stacks for the year 2019 based on DEHSt.

Of course, quantitative comparisons require that complete cross-sections of the plume could be examined. Whilst we measured mostly background concentrations at the horizontal edges of our flight pattern, we measured some significant $CO_2$

enhancements at the vertical edges, especially at the top in a height of 200 m, which was the maximum altitude allowed by flight regulations for that day. This strongly indicates that we have only seen parts of the plume and we assume that most of the plume has risen above the maximum flight altitude. Additionally, our result represents only a snapshot of the current situation which does not necessarily agree with the annual average emissions. Ideally, emissions should be compared only on the basis of instantaneous or short term average values.

Our uncertainty estimate is not affected by the fact that our measurements sampled only parts of the plume. It suggests that it is possible to determine the cross-sectional flux of a point source with this type of sUAS with an uncertainty of about 8.5% when considering only instrumental effects and neglecting, e.g., the influence of turbulence. The flux uncertainty is dominated by the uncertainty of the wind speed and valid for average wind speeds of about $4\,\mathrm{m\,s^{-1}}$. It is relatively insensitive to the magnitude of the flux and will reduce for moderately larger wind speeds, even though the $CO_2$ enhancements will become smaller. Our

uncertainty estimate is consistent with that of Krings et al. (2018) for in situ measurements taken aboard a manned aircraft in a larger distance to the source. They estimated the flux error due to the uncertainties of the primary measurements to be well below 10%.

It shall be noted that our uncertainty estimate only accounts for instrumental effects and we agree with Krings et al. (2018) that there are other factors which can result in significant differences between the inferred total flux and the facility's actual





405 emissions. These are, for example, turbulence (i.e., short term fluctuations due to non-steady meteorology) or undersampling of the plume morphology. However, such effects can average out when averaging over a large enough number of cross-sections, i.e., long enough flight times so that the uncertainty converges towards our estimate of systematic instrumental effects.

The introduced sUAS can also be used for weaker or stronger sources as long as a suitable distance to the source and extent of the flight pattern can be used. However, flight regulations often prescribe a minimum distance to be kept and, additionally,

410 flights are usually only permitted within visual line of sight, which limits the maximum extension of the flight pattern. One main target of future satellite missions such as CO2M is the quantification of anthropogenic $CO_2$ point source emissions and with the estimated accuracy, the sUAS is suitable to complement other, e.g., airplane based validation activities. Moreover it may also be useful for the sample verification of reported emissions and can provide emission estimates under conditions not suitable for satellite measurements, e.g., in cloud contaminated scenes or during night (if flight regulations allow).

415 However, it shall be noted that there are also several weather conditions under which flights with our sUAS are not possible or meaningful. For example strong winds may prevent a save operation, rain may destroy parts of the payload or prevent its functioning, and too low wind speeds may render the results hard to interpret because the mass flux becomes dominated by diffusion instead of advection. Beside this, many countries have put in place strict regulations for UAV flights (for good reasons), which also impacts the possible areas of application.

420 For the future, we envisage many interesting potential applications, improvements to be made, and scientific questions to be answered: A promising next step would be to use the sUAS for the quantification of the emissions of known sources by measuring complete plume cross-sections and investigate how much averaging has to be applied on the cross-sectional flux in order to converge to the actual emissions given short term fluctuations due to turbulence and potential undersampling of the complex plume morphology. As the wind measurement is the dominating source of uncertainty, it should be investigated how

425 these measurements can be further improved. This could, e.g., include experiments with an anemometer with reduced noise or the implementation of more complex calibration models. In addition, the capabilities of the onboard computer should be exploited to allow fully autonomous flights with a flight pattern automatically adapted to the wind and $CO_2$ measurements, resume after battery replacement, complex no-fly zones, etc. The change to a more advanced UAV could enable longer flight times, heavier payloads (e.g., for simultaneous measurements of other gases), and flights at higher wind speeds.



430 **Appendix A**

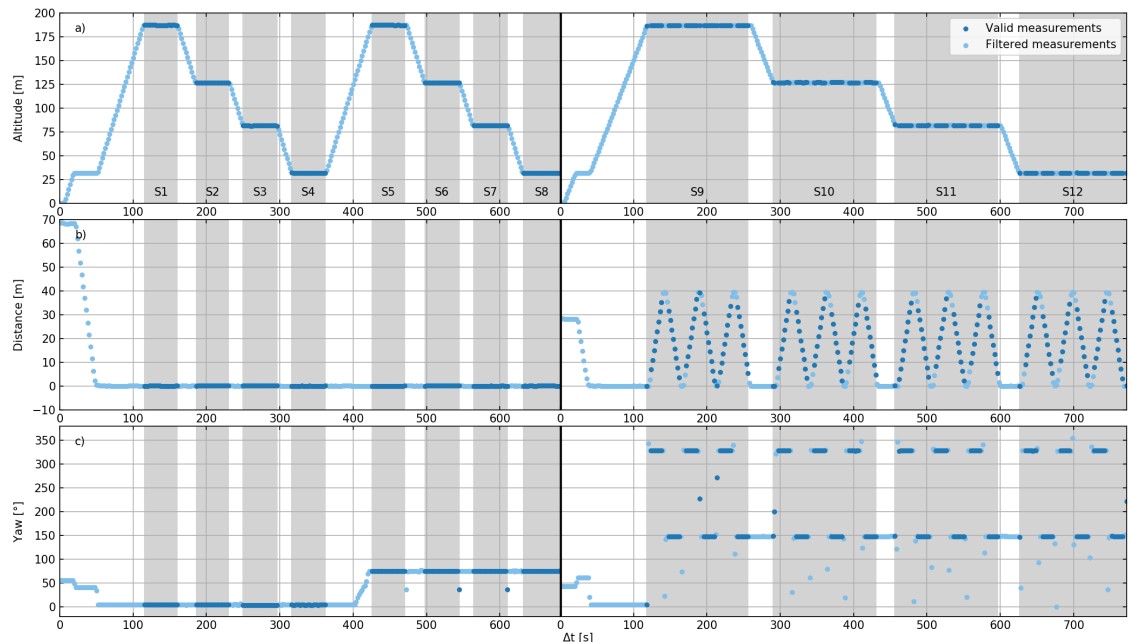

**Figure A1.** Detailed flight pattern of the validation flights at the ICOS atmospheric station Steinkimmen (STE), Germany, 09.04.2020. From top to bottom: altitude (**a**), horizontal distance from the first valid measurement within each flight (**b**), yaw of the UAV (**c**). **Left**: flight 1. **Right**: flight 2.





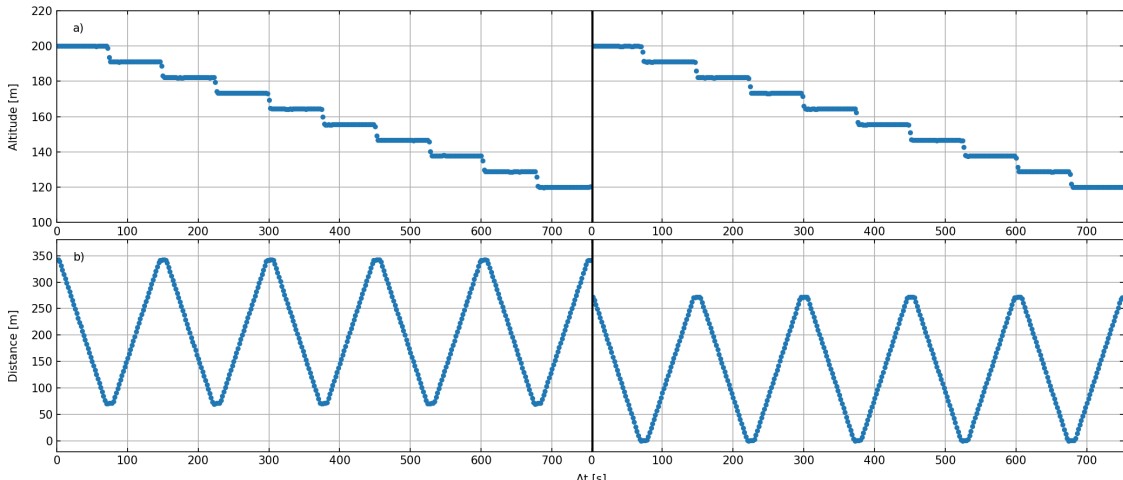

**Figure A2.** Detailed flight pattern of two flights downwind of the ExxonMobil natural gas processing facility in Großenkneten 40 km east of Bremen, Germany, 10.04.2020. **a)** Altitude. **b)** Horizontal distance from the west-most point of the flight pattern. **Left**: flight 1. **Right**: flight 2.

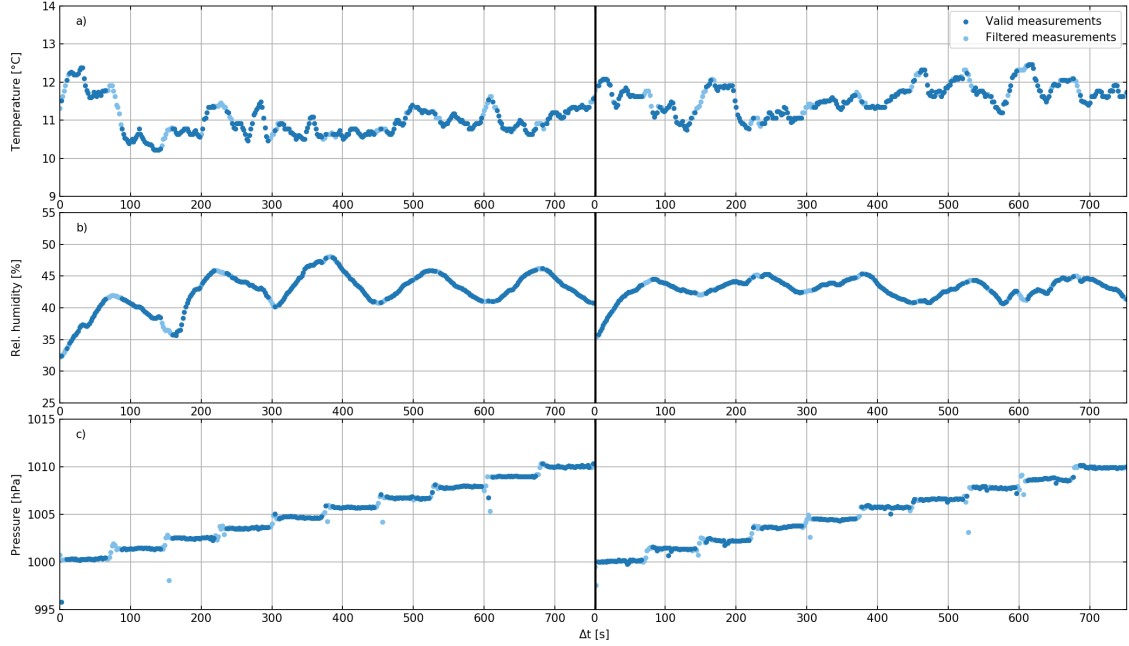

**Figure A3.** Temperature (**a**), relative humidity (**b**), and pressure measured (**c**) during two flights downwind of the ExxonMobil natural gas processing facility in Großenkneten 40 km east of Bremen, Germany, 10.04.2020. **Left**: flight 1. **Right**: flight 2.



*Data availability.* The shown measurement data can be made available on request.

*Author contributions.* M.R.: design and operation of the sUAS, experimental set-up, data analysis, interpretation, writing the paper. M.B.: assistence in target selection, interpretation, improving the paper. H.B.: assistence in target selection and conducting the flights, interpretation, improving the paper. J.B., S.K. and K.G.: assistence in the comparison with the LGR instrument, improving the paper. M.L. and D.K.:
435  provision and assistence in the interpretation of the ICOS data, improving the paper. J.P.B.: interpretation, improving the paper.

*Competing interests.* The authors declare that they have no conflict of interest.

*Acknowledgements.* This work has been funded by the State and the University of Bremen. The ICOS observations at Steinkimmen were conducted within the ICOS RI consortium by scientists contributing to the different components (National Networks, Central Facilities and Carbon Portal) that are jointly funded by national funding agencies from all ICOS partner countries.



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
