# Peer review of "Development of a small unmanned aircraft system to derive CO2 emissions of anthropogenic point sources"

_Atmospheric Measurement Techniques, 2020_

## Referee Comment (RC1) · Anonymous Referee #1 · 9 Oct 2020

**Review of "Development of a small unmanned aircraft system to derive CO2 emissions of anthropogenic point sources", *Atmospheric Measurement and Techniques*, by Maximillian Reuter et al., doi.org/10.5194/amt-2020-234**

**Anonymous Reviewer**

This study outlines the development of a small unmanned aircraft system (sUAS) to measure  $CO_2$  emission point sources, consisting of a drone platform hosting a non-dispersive infrared (NDIR)  $CO_2$  sensor, a 2D ultrasonic anemometer, and pressure, temperature, and humidity sensors. Combined, this allows the system to take in-situ measurements of the mole fraction of  $CO_2$  at various heights, with associated micrometeorological measurements. The development of drone based atmospheric sensors fills an important niche between ground based and airplane-based measurements, particularly for measuring point sources in areas without established ground based sensors. As such, this study is pertinent for these developments.

Overall, the manuscript is very well prepared. The outline of the development of the drone based platform is well presented, and the authors have shown the extensive steps they have taken to ensure that the micrometeorological sensors on the platform are correctly synced to the measurements of the  $CO_2$  sensor. The authors have developed a robust correction for the  $CO_2$  sensor for the flight conditions which they expect the sensor to operate, and have also developed a means to account for rotor downwash on their 2D ultrasonic anemometer measurements. The authors have taken the necessary steps to test these corrections via an intercomparison study with  $CO_2$  and meteorological sensors at an ICOS station. The details provided through these stages are excellent, and consequently, I am convinced of the suitability of this sUAS for in-situ measurements, at height, of the mole fraction of  $CO_2$ . However, in order to be convinced of the suitability of the sUAS to derive emissions of  $CO_2$  from anthropogenic point sources, there are some areas which could be elaborated upon or further clarified with regards to the field campaign as outlined in Section 6 of the manuscript. I would encourage the manuscript to be published, once these issues are addressed:

- 1. As outlined in the paragraph beginning Line 341, the authors note that the inferred total flux is significantly lower than the facility's own estimate, and suggest that the discrepancy is due to the fact that only part of the plume was sampled. Line 291 clarifies that the legal operational parameters for the drone flights allowed for a maximum height of 200 m. The authors assessment is therefore highly likely to be correct. This raises the question of how suitable the sUAS is, as a self-contained platform, for inferring anthropogenic point emissions. The authors do note this in Line 388 ("Of course, quantitative comparisons require that complete cross-sections of the plume be examined"). The authors go on to suggest in Line 421-423 that future studies with the sUAS could seek to investigate "how much averaging has to be applied on the cross sectional flux in order to converge on the actual emissions". Would this not, however, require knowledge of the morphology of the plume? If so, then plume simulations would be required. I think it would be good to see in this study at least some modelling of the plume stack, such as using SHRMC-4S DAYSMOKE. I think inclusion of this would highlight the limitation of this sUAS for sampling plumes from industrial stacks, while concurrently explaining the difference between the study's inferred  $CO_2$  flux and the value taken from the facility. By doing this, it can be argued the sUAS is indeed suitable for point emissions measurements, but that in this example, legal limitations on drone operation prevented a full analysis. If it is not possible for the authors to do this, then the authors should include further technical argument in Section 6 and in the Summary regarding how emission sources could be inferred from sampling cross sections of the plume (perhaps outlining the steps that the authors would have taken if plume morphology was known).
- 2. Following from this point, an outline of the operating parameters of the sUAS, such as maximum allowed height of operation, maximum distance from observer, battery temperature etc., would be welcome somewhere in Section 2. The reader would benefit in understanding the limits to drone operation, particularly with regards to sampling heights. It is also important to clarify why this platform can operate at a 200 m flying height. Current EU legislation limits drones for both recreational and commercial flights to 120 m above surface level, so I'm a little confused as to why this sUAS was able to operate at 200 m! By adding this information, the reader would also be able to gauge whether this sUAS meets the requirements for their field site. Furthermore, it would also clarify some issues with turbulence. If the drone can only operate below a certain wind speed, then turbulence effects become less significant.
- 3. The x-axis on **Figure 9** is labelled "Distance [m]". It is important to clarify what this distance is from. Presumably, it is the distance from the attitude of the drone when measurements were first taken?

4. CO2 emissions from point sources are often associated with emissions of particulate matter. Have the authors considered the suitability of their sUAS to take downwind measurements of CO2 in plumes that may contain PM? How would drone and sensor performance be affected?

Beside these points concerning the content of the work, the following stylistic points are suggested:

- The axis labels of all figures are quite small and difficult to read without zooming in. Could the authors enlarge the x-axis and y-axis labels for all figures?
- For Figure 2, a secondary y-axis plot could be used to plot pressure on the same sub-plot as RH.

---

## Referee Comment (RC2) · Anonymous Referee #2 · 22 Oct 2020

**Review of **Development of a small unmanned aircraft system to derive CO2 emis**sions of anthropogenic point sources by Maximilian Reuter et al.**

The authors describe the design and a few test flights of a UAV to estimate CO2 emissions from a point source by an integral advection method. The paper is well written and the description of the UAV is comprehensive. The estimation of the accuracy for emission assessment is somewhat short but for a first guess acceptable. For a reliable application more flights are necessary. Furthermore, as anthropogenic point sources and regulated airspaces typically coincide the issue of airspace clearance for potential applications should be discussed.

A few specific comments:

Generally, all figure labels are too small. Figures should be redrawn before final publication. Some figures (e.g. Fig.4) would benefit from rescaling or splitting.

The anemometer calibration (Section 4) looks reasonable. With a small, slow aircraft that even can hold a fixed position, a comparison with a proven reference at a mast is possible, a big advantage to faster, manned planes. For future applications that characteristic should be exploited more in calibration and measurement pattern design.

Section 5, validation with ICOS: Why do you use level 1 data for comparison ? At low level there is more horizontal variation on a small scale in the flow and in the scalar fields than there is at higher level, where a horizontal distance to the reference has less effect.

p14, l240: The discrepancy in the wind data between UAV and mast around 700s (S12) looks like a problem in the directional data of the UAV. Can you comment on that ?

Has S12 been flown at 32 m? The height of the legs should be given.

You could even try to get the mean wind by having the UAV drift with it by keeping only a constant height and horizontal leveling active and deactivate position holding. Then, the drift speed should be the mean wind speed, like with a radiosonde. A tilt error should then be checked by repeating with the UAV turned by 180deg around the vertical axis. Minor specific point, typos and such:

- p3, l85: ... via an RS-232, an RS-485 ...
- p4, caption Fig.1: the labels in the left photography of are hard to read
- p6, Fig.2: labels far too small, caption too brief
- p9, Fig.3: labels too small
- p10, Fig.4: labels too small, abscissa could be reduced to the range of 350s–1200s, e,f,g: labelling/text mismatch.
- p12, Fig.5: labels too small
- p13, Fig.6: labels too small.
- p15, l266: A **?possible?** explanation
- p15, l276: 40k **west** of Bremen, dito in captions Fig.7 and 8., p20,l382
- p16, l305: while this method seems acceptable for a first estimate in using a new device, shouldn't the background concentration be determined on the upwind side of the emitter ?
- p17, l308: With a linear interpolation you assume a horizontal gradient in the concentration, why ? Why dont't you take the average ? Especially as you assume no upwind sources anyway.
- p19, l339: ... beyond the scope of this paper ...: Well, you could at least tell us the difference in the estimate between both flights.

---

## Author Comment (AC1) · 5 Nov 2020

First of all, we thank reviewer 1 for his effort in carefully reviewing our manuscript and his constructive comments.

**Point-by-point answers to the comments of reviewer 1**

*Reviewer 1:* *As outlined in the paragraph beginning Line 341, the authors note that the inferred total flux is significantly lower than the facility's own estimate, and suggest that the discrepancy is due to the fact that only part of the plume was sampled. Line 291 clarifies that the legal operational parameters for the drone flights allowed for a maximum height of 200 m. The authors assessment is therefore highly likely to be correct. This raises the question of how suitable the sUAS is, as a self-contained platform, for inferring anthropogenic point emissions.*

**Authors:** Technically, the used DJI UAV can fly much higher but its software limits the maximum altitude to 500 m above ground. Whether 500 m is sufficient to fully sample the entire plume structure depends on many factors such as stack altitude, atmospheric stability, distance to the source, exit temperature and velocity of the flue gas, wind speed, etc. However, especially when flying in relatively stable atmospheric conditions, only a couple of hundred meters downwind of the source, 500 m will most likely often be sufficient.

Whether such flights are legally allowed is of course another question that cannot be answered so easily in general. The legal regulations vary significantly from country to country and are subject to frequent adjustments and they often also depend on which organization is responsible for the flights. For example, some government agencies can have far-reaching permissions. In addition, the local aviation authority may grant special permissions, as was the case with our flights. We added this discussion of the legal aspects to Sec. 7 (Summary and Conclusions).

*Reviewer 1:* *The authors go on to suggest in Line 421-423 that future studies with the sUAS could seek to investigate "how much averaging has to be applied on the cross sectional flux in order to converge on the actual emissions". Would this not, however, require knowledge of the morphology of the plume?*

**Authors:** Of course, this will depend on the meteorological situation. Turbulent conditions will require more averaging than stable conditions.

*Reviewer 1:* *I think it would be good to see in this study at least some modelling of the plume stack, such as using SHRMC-4S DAYSMOKE. I think inclusion of this would highlight the limitation of this sUAS for sampling plumes from industrial stacks, while concurrently explaining the difference between the study's inferred CO2 flux and the value taken from the facility. By doing this, it can be argued the sUAS is indeed suitable for point emissions measurements, but that in this example, legal limitations on drone operation prevented a full analysis.*

**Authors:** We added the following discussion to Sec. 6 of the manuscript: 'We use a Gaussian plume model (Beychok, 2005) to estimate the expected plume

extend for moderately unstable conditions (Pasquill stability class B) resulting in a full width half maximum of 197 m horizontally and 124 m vertically. The expected corresponding plume rise can be estimated with Briggs' equations for bent-over, hot buoyant plumes (Beychok, 2005). Most input parameters to the Briggs equations such as temperature and wind speed at stack height have been measured but other parameters require ad hoc assumptions. We assume that the exiting flue gas consists of 21% $CO_2$ and that it is 50° warmer than the ambient air. By applying the ideal gas law, these values are used to estimate that the annual emissions through the main stacks have an average volumetric flow rate of roughly $78 \, \mathrm{m}^3 \, \mathrm{s}^{-1}$. For this scenario, Briggs' equations estimate that the expected center of the plume 500 m downwind of the source has risen to 234 m.

In case of a Gaussian plume morphology, this would mean that roughly 74% of the emitted $CO_2$ has risen above 200 m. However, it shall be noted that the Gaussian plume shape and a plume rise according to Briggs' equations is only on average a good estimate for reality but on short time scales, turbulence can result in large deviations form that. Also, the results of our simple simulations are relatively sensitive to the made ad hoc assumptions. Nevertheless, they indicate that the width of the flight pattern (about 340 m) was sufficient to sample the expected plume width, but that large parts of the plume may indeed have risen above the maximum flight altitude."

**Reviewer 1:** *An outline of the operating parameters of the sUAS, such as maximum allowed height of operation, maximum distance from observer, battery temperature etc., would be welcome somewhere in Section 2.*
**Authors:** As discussed above, the maximum allowed height depends on the legal permissions but we added some technical specifications taken from the fact sheet of the UAV to Sec. 2: "According to the technical specifications of the UAV, it can operate in temperatures between -20°C and 50°C; its maximum flight altitude is reached at 3 km above mean sea level (with special propellers) or at 500 m above ground level (limited by its firmware); under optimal conditions, its radio system can operate over distances of up to 8 km; its maximum wind resistance is $12 \, \mathrm{m} \, \mathrm{s}^{-1}$."

**Reviewer 1:** *It is also important to clarify why this platform can operate at a 200 m flying height. Current EU legislation limits drones for both recreational and commercial flights to 120 m above surface level.*
**Authors:** We got a special permission from the local aviation authority (see discussion above).

**Reviewer 1:** *The x-axis on Figure 9 is labelled "Distance [m]". It is important to clarify what this distance is from. Presumably, it is the distance from the attitude of the drone when measurements were first taken?*
**Authors:** We added to the caption of Fig. 9: "The x-axis corresponds to the distance from the west-most point of the flight pattern."

***Reviewer 1:*** *CO2 emissions from point sources are often associated with emissions of particulate matter. Have the authors considered the suitability of their sUAS to take downwind measurements of CO2 in plumes that may contain PM? How would drone and sensor performance be affected?*

**Authors:** The DJI Matrice 210v2 is advertised to feature a rugged design optimized for industrial applications and has a IP43 Ingress Protection Code. Therefore, we do not expect any problems with moderate concentrations of co-emitted PM.

The Vaisala GMP343 NDIR $CO_2$ sensor analyzes reflected infrared light at two wavelengths around 4µm. In principle, scattering at aerosols within the sensor's cavity can modify (usually reduce) the light path which would be misinterpreted as change in atmospheric $CO_2$. However, at these wavelengths aerosol extinction is usually far less pronounced compared to the visible spectral region. Additionally, we expect that mainly soot aerosols are co-emitted having a small single scattering albedo. This means, most of the radiance extinction is due to absorption but not scattering. This (spectrally broad-band) absorption along the light path will not significantly influence the sensor results because the measurement principle uses two neighboring wavelengths from which one serves as reference. Therefore, we consider co-emitted PM to be not a significant error source for our flux estimation whose uncertainty budget is dominated by the uncertainty of the wind measurements (8.3%).

***Reviewer 1:*** *The axis labels of all figures are quite small and difficult to read without zooming in. Could the authors enlarge the x-axis and y-axis labels for all figures?*

**Authors:** Done. We enlarged the font size of almost the entire text shown in the figures.

***Reviewer 1:*** *For Figure 2, a secondary y-axis plot could be used to plot pressure on the same sub-plot as RH.*

**Authors:** If OK for reviewer 1, we would prefer to keep the simple, clear design of the figure.

**References**

Beychok, M. R.: Fundamentals Of Stack Gas Dispersion (4th ed.), author-published, 2005.

---

## Author Comment (AC2) · 5 Nov 2020

First of all, we thank reviewer 2 for his effort in carefully reviewing our manuscript and his constructive comments.

**Point-by-point answers to the comments of reviewer 2**

**Specific comments**

**Reviewer 2:** Generally, all figure labels are too small. Figures should be redrawn before final publication. Some figures (e.g. Fig.4) would benefit from rescaling or splitting.

Authors: Done. We enlarged the font size of almost the entire text shown in the figures. The abscissa in Fig. 3 and 4 has been re-scaled.

**Reviewer 2:** The anemometer calibration (Section 4) looks reasonable. With a small, slow aircraft that even can hold a fixed position, a comparison with a proven reference at a mast is possible, a big advantage to faster, manned planes. For future applications that characteristic should be exploited more in calibration and measurement pattern design.

**Authors:** Our comparison of the calibrated wind measurements with mast-based reference measurements presented in Sec. 5 (Validation using ICOS measurements) indeed shows that the UAS derived wind agrees well with reference data. However, given that the uncertainty of emission estimates are often driven by the uncertainty of the wind measurements, we intend to continue our efforts to monitor and ideally also to improve the quality of our calibration in the future, if opportunities arise.

**Reviewer 2:** Section 5, validation with ICOS: Why do you use level 1 data for comparison ? At low level there is more horizontal variation on a small scale in the flow and in the scalar fields than there is at higher level, where a horizontal distance to the reference has less effect.

**Authors:** We intended to capture as much vertical atmospheric variability (potential gradients in temperature, wind, humidity, and  $CO_2$ ) as possible. However, we agree that the deviations between our measurements and those performed at the mast may be larger for the lowermost measurement level compared to the higher levels. Most likely, this is the reason for the reduced agreement between our and the ICOS wind measurements in sequence S12 (see Fig. 6). We discuss this point in Sec. 5 (Validation using ICOS measurements) as follows: "The relative large scatter of the difference to ICOS is mainly driven by a poor agreement in S12. As visible in Fig. 5 (a and b), the tower is located close to a small piece of forest while the flight track is above a relatively free field which can lead to significant differences of the measurements at 32m height." This is also the reason, why we excluded S12 from the computation of the scatter of the wind components ("The average of the scatter of the north and east component is  $0.62 \,\mathrm{m\,s^{-1}}$  (excluding S12 ...).").

**Reviewer 2:** p14, l240: The discrepancy in the wind data between UAV and mast around 700s (S12) looks like a problem in the directional data of the UAV. Can you comment on that ?

Authors: As discussed above, we expect, that the poor agreement of the wind measurements in S12 (at 32m) comes from the influence of the surface.

**Reviewer 2:** Has S12 been flown at 32m? The height of the legs should be given.

**Authors:** Yes, S12 is at 32m. We mention this in L241. Additionally, we describe the flight pattern in Sec. 5 (Validation using ICOS measurements): "During the first flight, the UAV twice ascended to 187m and descended level by level to 32m ... The flight pattern is shown in detail in Fig. A1.". The referenced figure A1 shows that S12 corresponds to 32m.

**Reviewer 2:** You could even try to get the mean wind by having the UAV drift with it by keeping only a constant height and horizontal leveling active and deactivate position holding. Then, the drift speed should be the mean wind speed, like with a radiosonde. A tilt error should then be checked by repeating with the UAV turned by 180deg around the vertical axis.

**Authors:** We agree, this could indeed be an interesting additional analysis for future calibration and validation flights.

**Minor specific point, typos and such**

**Reviewer 2:** p3, 185: ... via an RS-232, an RS-485 ... Authors: Done.

**Reviewer 2:** p4, caption Fig.1: the labels in the left photography of are hard to read

Authors: We experimented with different colors (red/white), with or without outlines, bold or normal fonts but did not find a better combination than the one currently selected.

**Reviewer 2: p6, Fig.2: labels far too small, caption too brief**

Authors: Done. The caption now reads: "a)  $CO_2$  concentration measured with the Vaisala GMP343  $CO_2$  sonde with and without linear correction (Eq. 1) as well as highly accurate reference measurements performed with an ABB LGR-ICOS ultra-portable greenhouse gas analyzer. b) Difference between the Vaisala GMP343  $CO_2$  (with and without linear correction) and the reference measurements. Pale colors represent instantaneous differences and intense colors 1h running averages. c) Temperature measured with the Vaisala GMP343. d) Relative humidity measured with a Sensirion SHT31-DIS sensor. e) Pressure measured with a Bosch BMP388 sensor. " **Reviewer 2:** p9, Fig.3: labels too small **Authors:** Done.

**Reviewer 2:** p10, Fig.4: labels too small, abscissa could be reduced to the range of 350s-1200s, e,f,g: labelling/text mismatch. **Authors:** Done.

**Reviewer 2:** p12, Fig.5: labels too small **Authors:** Done.

**Reviewer 2:** p13, Fig.6: labels too small. **Authors:** Done.

**Reviewer 2:** p15, l266: A ?possible? explanation **Authors:** In L269ff we discuss: "A potential explanation of the general overestimation could be the heating of the GMP343 CO2 sonde, which is intended to reduce the possibility of condensation on the optical components but which could also slightly warm the temperature sensor."

**Reviewer 2:** p15, l276: 40k west of Bremen, dito in captions Fig.7 and 8., p20,l382 **Authors:** Done.

**Reviewer 2:** p16, l305: while this method seems acceptable for a first estimate in using a new device, shouldn't the background concentration be determined on the upwind side of the emitter ?

Authors: According to Gauss's theorem, one would ideally have continuous measurements all around and above the facility to exactly quantify the inflow of  $CO_2$ . Of course, this is not possible so that an as good as possible choice of the undisturbed background concentration has to be made. Upwind measurements would indeed minimize the influence of potential upwind sources. However, they would have the disadvantage, that they cannot be performed during the same flight because of the larger distance. Furthermore, the flight time in the background air would considerably be extended at the expense of the flight time within the plume. For these reasons, we considered nearly simultaneous measurements in the left and right neighborhood of the plume as better choice to estimate the undisturbed background concentration.

Please also note that similar approaches are often used in the literature (e.g., Carotenuto et al., 2018; Krings et al., 2018) and that Krings et al. (2018) discuss: "The single-screen approach was chosen for practical reasons, because flying around a source means spending most of the time in background concentrations ... Some circumferential tracks ... confirmed the background concentrations found on the edges of the single screens."

Reviewer 2: p17, l308: With a linear interpolation you assume a

horizontal gradient in the concentration, why ? Why dont't you take the average ? Especially as you assume no upwind sources anyway.

**Authors:** As one can see in Fig. 8, there is no significant variability in the background concentrations. Therefore, it would practically make no difference to use the average or the linear interpolation. However, we consider the linear interpolation a good choice because theoretically, the background may vary slightly along the flight path, e.g., because of upwind variations of natural fluxes. Additionally, small sensor drifts may exist, which would also be better accounted for by a linear interpolation.

**Reviewer 2:** p19, l339: ... beyond the scope of this paper ...: Well, you could at least tell us the difference in the estimate between both flights.

**Authors:** As shown in Fig. 7, we shifted the flight pattern of both flights against each other, so that the sampling becomes denser in the center. The downside of this strategy is, that the western pattern likely misses some parts of the plume in the east and the eastern pattern some parts in the west (see Fig. 7). Therefore, the expected values of the emissions derived from the individual flights are not identical to the expected value of the combined flight. For this reason, we decided to omit the discussion of the individual results in the paper and would prefer to keep it as is.

Nevertheless, out of curiosity, we computed the cross-sectional fluxes per flight and found  $209\pm17 \,\mathrm{ktCO_2} \,\mathrm{yr^{-1}}$  for the eastern and  $156\pm17 \,\mathrm{ktCO_2} \,\mathrm{yr^{-1}}$  for the western flight pattern. The uncertainty estimates have been computed in the same way as done for the combined cross-sectional flux. As discussed in the paper, this uncertainty estimate is incomplete. Foremost it does not consider turbulence which is one (or even the main) reason why it is no contradiction that both values differ by more than their uncertainties.

**References**

- Carotenuto, F., Gualtieri, G., Miglietta, F., Riccio, A., Toscano, P., Wohlfahrt, G., and Gioli, B.: Industrial point source CO2 emission strength estimation with aircraft measurements and dispersion modelling, Environmental monitoring and assessment, 190, 165, 2018.
- Т., Neininger, В., Gerilowski, Κ., Krautwurst, S., Krings, Buchwitz, М., Burrows, J. Р., Lindemann, С., Ruhtz, Т., Schüttemeyer, D., and Bovensmann, H.: Airborne remote sensing measurements of atmospheric and in situ  $CO_2$  $\operatorname{to}$ quantify point source emissions, Atmospheric Measurement Techniques, 11. 721 - 739.https://doi.org/10.5194/amt-11-721-2018, URL https://www.atmos-meas-tech.net/11/721/2018/, 2018.